# SEMANTIC RE-TUNING WITH CONTRASTIVE TENSION

**Fredrik Carlsson**[*]     **Evangelia Gogoulou**     **Erik Ylipää**
**Amaru Cuba Gyllensten**     **Magnus Sahlgren**
RISE - NLU Group
{firstname.lastname}@ri.se

## ABSTRACT

Extracting semantically useful natural language sentence representations from pre-trained deep neural networks such as Transformers remains a challenge. We first demonstrate that pre-training objectives impose a significant task bias onto the final layers of models, with a layer-wise survey of the Semantic Textual Similarity (STS) correlations for multiple common Transformer language models. We then propose a new self-supervised method called Contrastive Tension (CT) to counter such biases. CT frames the training objective as a noise-contrastive task between the final layer representations of two independent models, in turn making the final layer representations suitable for feature extraction. Results from multiple common unsupervised and supervised STS tasks indicate that CT outperforms previous State Of The Art (SOTA), and when combining CT with supervised data we improve upon previous SOTA results with large margins.

## 1 INTRODUCTION

Representation learning concerns the pursuit of automatically learning representations of data that are useful for future extraction of information (Bengio et al., 2013). Recent work has predominantly been focused on training and extracting such representations from various deep neural architectures. However, as these deep models are mostly trained via error minimization of an objective function applied to the final layers (Rumelhart et al., 1988), features residing in layers close to the objective function will be task-specific Yosinski et al. (2014). Therefore, to reduce the representation's bias towards the objective function it is common to discard one or several of the final layers, or alternatively consider features of other intermediate layers, as with AutoEncoders (Rumelhart et al., 1986).

One domain where this issue is particularly striking is learning semantic sentence embeddings with deep Transformer networks (Vaswani et al., 2017) pre-trained towards some language modeling task. Although utilizing pre-trained Transformer models such as BERT, XLnet, ELECTRA and GPT-2(Devlin et al., 2019; Yang et al., 2019; Clark et al., 2020; Brown et al., 2020) has become the dominant approach within the field of Natural Language Processing (NLP), with current State Of The Art (SOTA) results in basically all NLP tasks belonging to fine-tuned versions of such models, it has been shown that simply extracting features from the layers of such models does not produce competitive sentence embeddings (Reimers & Gurevych, 2019; Liu et al., 2019a). Our interpretation of this phenomenon, which we will demonstrate in this paper, is that the currently used language modeling objectives enforce a task-bias at the final layers of the Transformer, and that this bias is not beneficial for the learning of semantic sentence representations.

Reimers & Gurevych (2019) propose to solve this by pooling a fixed size sentence embedding from the final Transformer layer and fine-tune towards a Natural Language Inference (NLI) task, an approach that when applied to Transformers is known as Sentence-BERT (or S-BERT in short). While Hill et al. (2016a) empirically show that fine-tuning language models towards NLI data yields good results on Semantic Textual Similarity (STS), there exists no convincing argument for why NLI is preferred over other tasks. Hence, it is unclear whether the impressive improvements of S-BERT are to be mainly attributed to the NLI task itself, or if this merely trains the model to output sentence embeddings, in turn exposing the semantics learned during pre-training. Since NLI requires labeled data, it would be highly valuable if an alternative method that requires no such labels was possible.

---

[*]Main contribution.

We therefore propose a fully self-supervised training objective that aims to remove the bias posed by the pre-training objective and to encourage the model to output semantically useful sentence representations. Our method trains two separate language models on the task of maximizing the dot product between the two models' representations for identical sentences, and minimizing the dot product between the models' representations for different sentences. When applied to pre-trained BERT models, our method achieves SOTA results for multiple unsupervised STS tasks, and when applied to the S-BERT model it outperforms previous SOTA by a clear margin. To further bolster the robustness of our method, we demonstrate that CT drastically improves STS scores for various models, across multiple languages.

Additionally, we contribute with a layer-wise STS survey for the most common Transformer-based language models, in which we find great variability in performance between different architectures and pre-training objectives. Finally, by introducing an alteration to the supervised regression task of S-BERT, we are able to improve upon the supervised STS embedding results for all tested models. In summary, the main contributions of our paper are as follows:

1. A novel self-supervised approach for learning sentence embeddings from pre-trained language models.

2. Analytical results of the layer-wise STS performance for commonly used language models.

3. An improvement to the supervised regression task of S-BERT that yields a higher performance for all tested models.

Code and models is available at *Github.com/FreddeFrallan/Contrastive-Tension*

## 2 RELATED WORK

Where earlier work for learning sentence embeddings focused on the composition of pre-trained word embeddings (Le & Mikolov (2014); Wieting et al. (2015); Arora et al. (2016)), recent work has instead favored extracting features from deep neural networks. The training methods of such networks can be divided into supervised and self-supervised. A systematic comparison of pre-Transformer sentence embedding methods is available in the works of Hill et al. (2016b).

**Self-supervised** methods typically rely on the assumption that sentences sharing similar adjacent sentences, have similar meaning. Utilizing this assumption, Kiros et al. (2015) introduced Skip-Thoughts that trains an encoder-decoder to reconstruct surrounding sentences from an encoded passage. Logeswaran & Lee (2018) proposed QuickThoughts that instead frames the training objective as a sentence context classification task. Recently, and still under peer-review, Giorgi et al. (2020) proposed DeCLUTR that uses a setup similar to QuickThoughts, but allow positive sentences to be overlapping or subsuming (one being a subsequence of the other), which further improves results.

**Supervised** methods utilize labeled datasets to introduce a semantic learning signal. As the amount of explicitly labeled STS data is very limited, supervised methods often rely on various proxy tasks where more labeled data is available. Conneau et al. (2017) introduced InferSent that learns sentence embeddings via a siamese BiLSTM trained on NLI data. The Universal Sentence Encoder (USE) of Cer et al. (2018) is a Transformer encoder trained with both unlabeled data and labeled NLI data. S-BERT by Reimers & Gurevych (2019) adopts the training objective of InferSent but instead applies pre-trained BERT models. Finally, Wang & Kuo (2020) recently proposed S-BERT-WK, an extension to S-BERT that further increases the performance by subspace analysis of the model's layer-wise word features.

Recently, Grill et al. (2020) introduced the self-supervised BYOL framework that attain useful image representations, comparable with previous supervised methods. Although their method also utilizes two untied dual networks, the main training objective and the underlying motivation for this differ greatly. Where BYOL train using solely positive samples generated via data augmentation, our method mainly aims to dissipate negative examples and relies on two networks in order to stabilize the training process. To the best of our knowledge, our work is the first that suggests learning sentence representations by removing the bias imposed from the pre-training objective.

# 3  LAYER-WISE STUDY OF TRANSFORMER MODELS

Previous work analyzing the downstream applicability of layer-wise features in Transformer model reports similar trends of performance increasing until the middle layers before decreasing towards the final layers. Merchant et al. (2020) found the best suited features for linguistic tasks such as entity typing and relation classification reside in the intermediate layers of BERT, and Chen et al. (2020) found the most useful representations for image classification in the intermediate layers of Image-GPT.

We contribute with a layer-wise study of the semantic quality of the sentence representations found in a selected number of common Transformer architectures. Following the approach of S-BERT, we generate sentence embeddings by mean pooling over the word-piece features of a given layer. These sentence embeddings are directly evaluated towards the STS-b test (Cer et al., 2017), without any additional training, from which we report the Spearman correlation between the cosine similarity of the embeddings and the manually collected similarity scores. The test partition of the dataset contains 1,379 sentence pairs, with decimal human similarity scores ranging from 0.0 (two sentences having completely different meanings) to 5.0 (two sentences have identical meaning).

Figure 1 shows the results for BERT, Electra, XLNet and GPT-2, with results for additional models in appendix B.4. Although the different models display different layer-wise patterns, a common theme is that it is not obvious where to extract features for semantic sentence embeddings; the worst-performing representations are often found in the layers close to the objective function, with the exception of RoBerta base (Liu et al., 2019b). Considering the discrepancy between BERT and Electra which share an almost identical architecture but differ drastically in their pre-training objectives, it is clear that the semantic quality of a model's sentence representations is heavily impacted by the choice of pre-training objective.

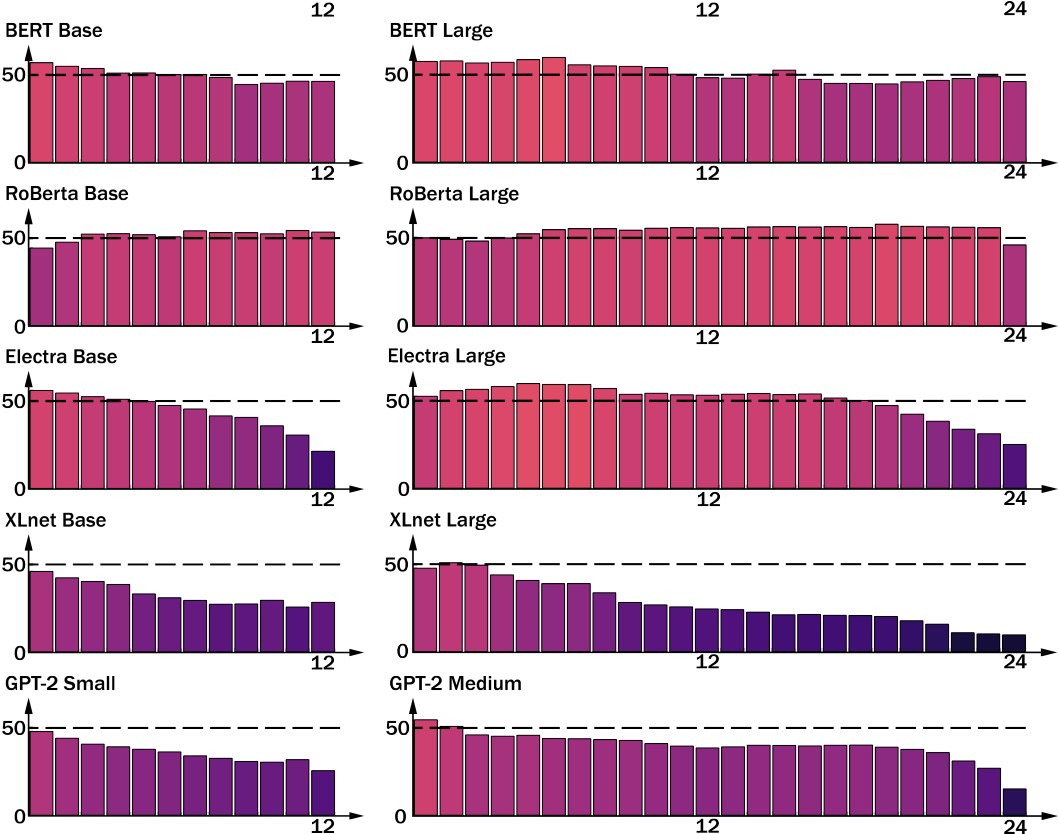

Figure 1: Layer-wise unsupervised STS performance on the STS-b test set. X-axis denotes the layers of the depicted model and the Y-axis denotes the Spearman correlation (x100). Color is a redundant indicator of the Spearman correlation and the value 50 is included for visual comparison.

## 4    METHOD

To counter the negative trend found in Section 3, where the lacking STS performance of the sentence representations in the final layers became apparent, we define a training objective meant to encourage the model to retain a semantically distinguishable sentence representation until the final layer. We name this method *Contrastive Tension* (CT), where two independent models, with identically initialized weights, are set to maximise the dot product between their sentence representations for identical sentences, and minimize the dot product for their sentence representations of differing sentences. Hence, the CT objective is defined as:

$$z = f_1(s_1)^T \cdot f_2(s_2)$$
$$\mathcal{L}(z, s_1, s_2) = \begin{cases} -log\ \sigma(z) & \text{if } s_1 = s_2 \\ -log\ \sigma(1-z) & \text{if } s_1 \neq s_2 \end{cases} \tag{1}$$

Where $f_1$ and $f_2$ are two independently parameterized models that given a sentence $s$ produces a fixed size vector representation and where $\sigma$ refers to the Logistic function.

Following the works of Reimers & Gurevych (2019), we generate fixed size sentence representations by mean pooling over the features in the final layer of pre-trained transformer models. Training data is randomly generated from a given corpus, where for each randomly selected sentence $s$, $K$ negative sentences are sampled to generate $K + 1$ training samples by pairing $s$ with the negative sentences and copying $s$ into an identical sentence pair. This yields one positive training sample and $K$ negative training samples. We include the $K + 1$ training samples in the same batch and always use $f_2$ to embed the $K$ negative sentences (See Appendix A.1 for a visual example). Our approach for generating negative samples is based on the assumption that two randomly selected sentences are very likely to be semantically dissimilar.

As the models are initialized with identical weights, the CT objective creates a tension between having the two models retain similar representations for identical sentences, at the same time as the two models are encouraged to distinguish their representations for differing sentences. Our intuition is that this creates a training dynamic where the two models acts as *smooth anchors* to each other, where the tension to remain synchronized mitigates the downsides of simply distancing the embeddings of differing sentences. This makes CT a nondestructive method for distinguishing the sentence embeddings of non semantically similar sentences.

## 5    EXPERIMENTS

Unless stated otherwise, the following set of hyperparameters is applied when using CT throughout all experiments: Training data is randomly sampled from English Wikipedia (See Appendix C.2), where we collect $K = 7$ negative sentence pairs for each positive sentence pair. The batch size is set to 16, which results in every batch having 2 positive sentence pairs and 14 negative sentence pairs. We apply an RMSProp optimizer (Hinton, 2012) with a fixed learning rate schedule that decreases from $1e^{-5}$ to $2e^{-6}$ (Appendix A.3). To showcase the robustness and unsupervised applicability of CT, we strictly perform 50,000 update steps before evaluating, and for all unsupervised tasks we report results for the **worst-performing** of the two models used in the CT setup. The experiment section follows the model naming convention elaborated upon in A.2, which describes the order and what training objectives that has been applied to a model.

There exists a clear discrepancy between previously reported STS scores for various methods and models. To improve upon this state of confusion we perform all evaluation with the SentEval package (Conneau & Kiela, 2018), to which we provide code and models for full reproducability of all tested methods. A Discussion regarding our experience with trying to reproduce previous work is available in Appendix A.4. A comprehensive list of all used model checkpoints is available in Appendix C.1

Table 1: Pearson and Spearman correlation (x100) on various unsupervised semantic textual similarity tasks.

| | STS12 | STS13 | STS14 | STS15 | STS16 | Avg. |
|---|---|---|---|---|---|---|
| InferSent-GloVe | 56.39 / 57.27 | 56.02 / 55.22 | 65.53 / 63.41 | 67.79 / 69.02 | 64.10 / 65.09 | 62.00 / 62.00 |
| USE v4 | 67.37 / 65,56 | 67.11 / 67.95 | 74.32 / 71.48 | 80.03 / 80.82 | 77.79 / 78.74 | 73.32 / 72.91 |
| BERT-Distil | 54.03 / 56.15 | 58.24 / 59.83 | 63.00 / 60.42 | 67.33 / 67.81 | 67.22 / 69.01 | 61.96 / 62.64 |
| BERT-Base | 46.88 / 50.07 | 52.77 / 52.91 | 57.15 / 54.91 | 63.47 / 63.37 | 64.51 / 64.96 | 56.96 / 57.24 |
| BERT-Large | 42.59 / 49.01 | 47.35 / 50.88 | 49.31 / 49.69 | 55.56 / 56.79 | 60.43 / 61.41 | 51.05 / 53.56 |
| S-BERT-Distil | 64.07 / 63.06 | 66.42 / 68.31 | 72.29 / 72.23 | 74.44 / 75.09 | 71.17 / 73.86 | 69.68 / 70.51 |
| S-BERT-Base | 66.61 / 63.80 | 67.54 / 69.34 | 73.22 / 72.94 | 74.34 / 75.16 | 70.16 / 73.27 | 70.37 / 70.90 |
| S-BERT-Large | 66.90 / 66.85 | 69.42 / 71.46 | 74.20 / 74.31 | 77.26 / 78.26 | 72.82 / 75.12 | 72.12 / 73.20 |
| S-BERT-Base-WK | 70.23 / 68.26 | 68.13 / 68.82 | 75.46 / 74.26 | 76.94 / 77.54 | 74.51 / 76.97 | 73.05 / 73.17 |
| S-BERT-Large-WK | 56.51 / 55.82 | 47.95 / 78.94 | 56.46 / 55.61 | 63.41 / 64.14 | 57.84 / 59.42 | 56.43 / 56.79 |
| *Our Contributions* | | | | | | |
| BERT-Distil-CT | 67.27 / 66.92 | 71.31 / 72.41 | 75.68 / 72.72 | 77.73 / 78.26 | 77.17 / 78.60 | 73.83 / 73.78 |
| BERT-Base-CT | 67.19 / 66.86 | 70.77 / 70.91 | 75.64 / 72.37 | 77.86 / 78.55 | 76.65 / 77.78 | 73.62 / 73.29 |
| BERT-Large-CT | 69.63 / 69.50 | 75.79 / 75.97 | 77.15 / 74.22 | 78.28 / 78.83 | **77.70 / 78.92** | 75.71 / 75.49 |
| S-BERT-Distil-CT | 69.39 / 68.38 | 74.83 / 75.15 | 78.04 / 75.94 | 78.98 / 80.06 | 74.91 / 77.57 | 75.23 / 75.42 |
| S-BERT-Base-CT | 68.58 / 68.80 | 73.61 / 74.58 | 78.15 / 76.62 | 78.60 / 79.72 | 75.01 / 77.14 | 74.79 / 75.37 |
| S-BERT-Large-CT | **71.70 / 69.80** | **73.95 / 75.45** | **78.10 / 76.47** | **80.39 / 81.34** | 75.93 / 78.11 | **76.01 / 76.23** |

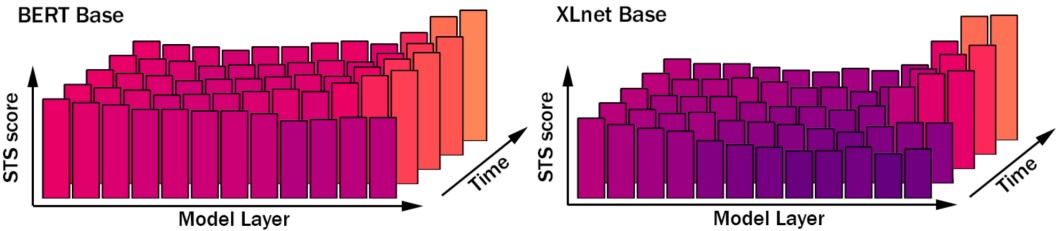

Figure 2: Layer-Wise STS performance on the STS-b test set throughout training with CT. X-axis depicts the layers of the model, Y-Axis depicts the Spearman correlation (x100) and the Z-Axis depicts the progression of time. Color is a redundant indicator of the Spearman correlation.

## 5.1 UNSUPERVISED STS

Table 1 shows the results for CT when evaluated on the unsupervised English STS tasks of Agirre et al. (2012; 2013; 2014; 2015; 2016). The non-fine-tuned BERT models perform the worst out of all considered models, with a decrease in performance as the size of the BERT models increase. CT clearly improves the results for all BERT based models and outperforms previous methods. When CT is applied to the supervised S-BERT models, it sets a new SOTA with a large margin (3.03 Spearman Points). Applying CT to BERT-Distil and S-BERT-Distil produces models that outperform S-BERT-Large while having 81% fewer parameters.

To further investigate the training dynamic of CT, we record the layer-wise STS performance for BERT and XLNet throughout the CT training process, by unsupervised evaluation on the STS-b test set. Figure 2 depicts the observed progression trends, and although the STS performance of the models' final layers differs greatly before fine-tuning, with XLNet showing drastically worse performance, both models clearly benefit from the CT training task. For both models, CT mainly affects the STS performance for the latter layers, as is to be expected from the low learning rate.

## 5.2 SUPERVISED STS

Reimers & Gurevych (2019) proposed an supervised STS regression task which directly targets the cosine similarity between sentence embeddings, creating regression labels by linearly mapping the human similarity scores to the range $[0, 1]$. However, as evaluation of STS related tasks uses the Pearson and Spearman correlation the range to which the cosine similarity labels are linearly mapped to is arbitrary. Hence, we propose to first investigate the spread within the models embedding space to find a model specific linear mapping of the regression labels that imposes less change to the current embedding space.

We investigate the STS spread of a model's embedding space by dividing the STS-b training data by their labels into 20 buckets, and measuring the mean cosine similarity between the sentence pairs within each respective bucket. As the STS-b data is labeled in the range $[0, 5]$, each bucket covers a range of $5/20 = 0.25$. Thus the lowest bucket contains all training samples with labels between $[0, 0.25]$ and the next bucket covers the range $(0.25, 0.5]$. The STS spread results for BERT and S-BERT before and after CT is available in Figure 3. We find that BERT produces sentence embeddings with high cosine similarity for all sentence pairs. Both CT and S-BERT improve the STS performance by decreasing the mean similarity for non-similar sentence pairs, but S-BERT does this with less precision.

After attaining prior knowledge about the model's sentence embedding space, we fine-tune towards the STS-b training data using the S-BERT regression setup, but with model specific regression labels. The cosine similarity labels are linearly mapped to the range $[M, 1]$, where $M$ is the mean cosine similarity of the lowest bucket. For each model and label scheme we perform 10 training runs for 8 epochs. Table 2 shows the test results of the model that performed best on the validation set.

We see a clear increase in performance for all models when utilizing the model specific regression labels. However, we find no significant increase in the supervised results when applying either CT, S-BERT, or a combination of the two prior to the supervised fine-tuning. As discussed in Appendix A.4 we failed to reproduce the results of Reimers & Gurevych (2019), which reports a mean Spearman correlation of S-BERT-Base: 85.35 and S-BERT-Large: 86.10, after training for 2 epochs.

Table 2: Pearson and Spearman correlation (x100) on the STS-b test set.

| Not trained for STS | | Trained with STS-b data | | |
|---|---|---|---|---|
| | | Regression Labels | [0, 1] | [M, 1] |
| BERT-Base | 47.91 / 47.29 | BERT-Distil | 84.07 / 84.23 | 85.02 / 85.54 |
| InferSent-GloVe | 65.30 / 63.21 | BERT-Base | 85.28 / 84.99 | 85.11 / 85.64 |
| USE v4 | 78.73 / 77.09 | BERT-Large | 85.54 / 85.37 | 85.90 / 86.35 |
| S-BERT-Distil | 73.88 / 76.19 | S-BERT-Distil | 84.22 / 84.26 | 85.40 / 85.64 |
| S-BERT-Base | 74.15 / 76.98 | S-BERT-Base | 85.17 / 84.90 | 85.59 / 85.81 |
| S-BERT-Large | 76.16 / 79.19 | S-BERT-Large | 85.14 / 85.07 | 85.25 / 86.28 |
| **Our contributions** | | | | |
| BERT-Distil-CT | 79.00 / 78.56 | BERT-Distil-CT | 84.14 / 84.19 | 85.32 / 85.82 |
| BERT-Base-CT | 77.87 / 76.32 | BERT-Base-CT | 85.13 / 84.92 | 85.76 / 85.89 |
| BERT-Large-CT | 79.97 / 78.99 | BERT-Large-CT | 85.20 / 84.97 | **86.37** / 85.89 |
| S-BERT-Base-CT | 76.25 / 80.11 | S-BERT-Distil-CT | 80.09 / 84.27 | 85.61 / 85.80 |
| S-BERT-Base-CT | 78.83 / 81.24 | S-BERT-Base-CT | 85.26 / 85.20 | 85.72 / 85.95 |
| S-BERT-Large-CT | **80.99 / 82.14** | S-BERT-Large-CT | 85.36 / 85.16 | 86.09 / **86.43** |

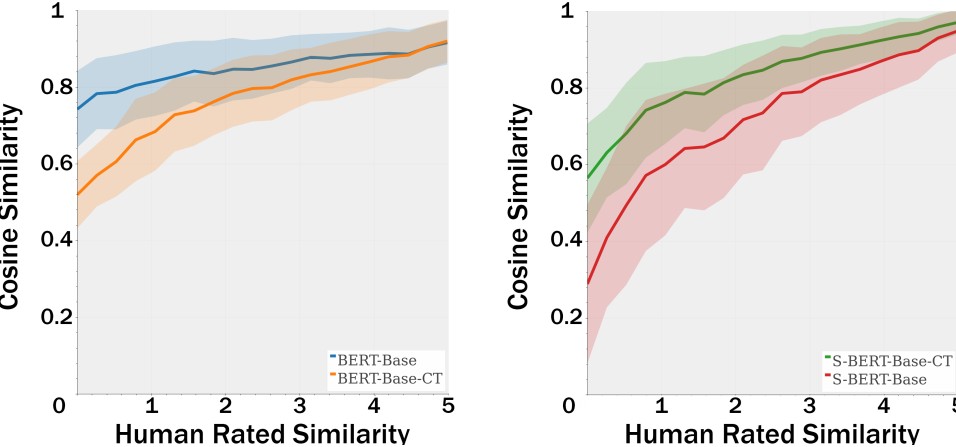

Figure 3: Predicted similarities for sentence pairs in the STS-b training set. Sentence pairs are chunked into 20 buckets by their labels, each bucket covering a label range of $0.25$. Opaque line denotes the mean and the transparent area denotes the standard deviation.

## 5.3 MULTILINGUAL STS

Table 3: Pearson / Spearman correlation (x100) on the STS test sets of various languages.

|  | Arabic | English | Russian | Spanish | Swedish |
|---|---|---|---|---|---|
| Native BERT | 37.92 / 45.21 | 47.91 / 47.29 | 64.34 / 65.75 | 67.41 / 69.19 | 41.90 / 44.91 |
| Multilingual BERT | 48.93 / 50.56 | 56.98 / 55.97 | 67.70 / 68.59 | 63.35 / 66.96 | 47.02 / 47.49 |
| XLMR | 46.64 / 44.76 | 40.54 / 40.35 | 60.93 / 61.28 | 57.30 / 59.31 | 42.16 / 42.03 |
| **Our Contributions** |  |  |  |  |  |
| Native BERT-CT | **67.91 / 67.57** | **77.87 / 76.32** | **79.38 / 79.62** | 76.02 / 76.07 | **61.69 / 61.68** |
| Multilingual BERT-CT | 60.12 / 60.15 | 64.39 / 62.28 | 70.14 / 70.54 | **77.67 / 77.96** | 58.63 / 57.45 |
| XLMR-CT | 62.30 / 62.14 | 69.85 / 68.22 | 67.96 / 68.42 | 76.00 / 77.30 | 59.29 / 58.19 |

We investigate the performance of CT when applied to various languages and evaluated towards STS data for Arabic, Spanish (Cer et al., 2017), Russian[1], and Swedish (Isbister & Sahlgren, 2020). All CT training is performed solely with data for the respective language towards which we evaluate it, using text from a Wikipedia dump of that language (See Appendix C.2). We perform experiments with three different types of pre-trained models, all of which have encountered the targeted evaluation language during pre-training: Native BERT models pre-trained with text data specifically for the targeted language, a multilingual BERT pre-trained for 104 languages and an XLM-R model pre-trained on 100 languages (Conneau et al., 2020)

Results in table 3 show that CT clearly improves the performance of all models. Prior to training with CT, we find that the multilingual BERT performs best, with the exception of Spanish where the native BERT performs the best. XLM-R performs the worst on all languages before CT. After training with CT the native BERT models outperform both the multilingual models, again with the exception of Spanish, where a slight edge is seen for the Multilingual BERT. The big performance increase seen on all models, on all languages, without requiring any labeled data, clearly demonstrate the robustness and unsupervised applicability of CT.

## 5.4 CORPUS VARIETY

To investigate how CT is impacted by different types of text data we vary the corpus from which training data is sampled. The corpora we consider are as follows: a dump of all English Wikipedia pages, a large book corpus comprised of $11,038$ books (Zhu et al., 2015), resulting in text data with a different tone and style compared to the text found on Wikipedia. Finally, we generate random word sequences by first uniformly sampling a sentence length in the range $[20, 75]$ and then filling that sequence with uniformly sampled tokens from the model's vocabulary.

For each corpus, we train 5 models with CT, and report the mean unsupervised Pearson and Spearman correlation on the STS-b test set. The results found in table 4 show that CT applied with different types corpus styles yields different STS results. All models attain their highest score with the Wikipedia data, with a noticeable performance drop with the book corpus and large performance drop using random data. Although the random corpus performs worst, it interestingly improves the performance of the smaller models while drastically worsening the performance of the large model. While the number of models used in this experiment is too small for conclusive evidence, the performance drop between corpus types seems correlated with model size.

Table 4: Unsupervised Pearson / Spearman correlation (x100) on the STS-b test set when performing CT with various corpora.

|  | Before CT | Wikipedia | Books | Random |
|---|---|---|---|---|
|  | *Pear / Spear* | *Pear / Spear* | *Pear / Spear* | *Pear / Spear* |
| Bert-Distil | 57.17 / 56.77 | **78.21 / 77.55** | 77.54 / 76.12 | 62.45 / 62.85 |
| Bert-Base | 47.91 / 47.29 | **75.30 / 73.75** | 73.30 / 70.95 | 52.95 / 52.42 |
| Bert-Large | 45.51 / 47.00 | **78.65 / 78.00** | 74.54 / 72.51 | 16.65 / 23.67 |

---

[1]https://github.com/deepmipt/deepPavlovEval

## 6 Discussion

The quality of a sentence representation depends on the generating model's ability to represent contextual interactions between the individual parts (in most cases, wordpieces) of the sentence. We refer to this as *compositionality*, for the lack of a better term. We propose that the task-bias of current Transformer language models can be seen as a form of compositionality-amnesia, where the models progressively express less compositional information throughout the layers, in favor of features specific to the pre-training objective. Sentence embedding methods attempt to correct for this bias by applying a learning criterion that enforces compositionality in the final layers.

In the case of CT, the learning objective is simply to maximize the dot product for identical sentences, and minimizing it for dissimilar ones. In the case of other techniques, the learning objective takes the form of modeling adjacent sentences (Skip-thoughts, Quick-thoughts, and DeCLUTR), or classifying entailment based on two given sentences (S-BERT), both of which have semantic interpretations. We argue that the CT objective is more suitable for the purpose of enforcing compositionality, since it only targets the composition function; there is no *semantics* involved in distinguishing identical from dissimilar sentences (all the necessary semantics is already learned by the language modeling objective).[2] The finding that CT works to some extent even with randomly generated sentences further strengthens this interpretation.

It is our intuition that CT is non-constructive, or in a sense uninformative: It does not add new information to the model, but rather forces the model to realign such that the compositional representation discriminates between different sentences. Hence, we find little reason to believe that the realignment enforced by CT to be beneficial for fine-tuning tasks where ample training data is available e.g. tasks for which a pre-trained BERT model can be fine-tuned with good performance. This is in accordance with the results available in Appendix 10, where the CT models are evaluated towards multiple supervised model tasks.

As can be seen in Figure 3, CT decreases the cosine-similarity for non-semantically similar sentences, while the cosine-similarity for highly semantically similar sentences mainly remain the same. We believe the reason for this desired behaviour to be that all representations are generated through a common parameterized compositionality function (the Transformer model). We thus find it unlikely that similar results could be attained by applying CT directly to individual representations so that the manipulation of individual embeddings is performed independently (as algorithms like Word2Vec does).

Our work demonstrates the potential to produce high-quality semantic sentence representations given a pre-trained Transformer Language model, without requiring any labeled data. In accordance with the multilingual results in Section 3, we would hence like to emphasize that this makes CT well suited for low resource languages, as neither pre-training or fine-tuning requires labeled data. Additionally, we think that interesting future work might consider exploring the possibilities of applying CT (or similar re-tuning objectives) during the pre-training of Transformer language model, and/or applying it to different intermediate layers

Finally, it is noteworthy that our results are not to be considered final, as many of the chosen hyperparameters for the CT experiments are yet to be thoroughly investigated. It is therefore possible that CT and similar methods can yield even better results, with either utilizing different language models or tuning of certain hyperparameters. Especially considering that during the unsupervised tasks, due to the emulation of having zero labeled data, we strictly performed a fixed number of iterations and assumed the worst-case scenario by reporting results for *the worst* performing model of the two CT models. A higher performance is therefore expected if the training is monitored with a validation set or if one were to combine the output of both CT models.

---

[2]This can easily be demonstrated by applying CT to a randomly initialized Transformer, which will not be able to learn anything useful; the semantics has to be already present in the model for CT to work.

## 7 CONCLUSION

This paper contributed with a layer-wise survey of the unsupervised STS performance of pre-trained Transformer language models. Results from this survey indicates that the final layer of most models produce the worst performing representations. To overcome this we proposed the self-supervised method Contrastive Tension (CT) that trains the model to output semantically distinguishable sentence representations, without requiring any labeled data. In an unsupervised setting CT achieves strong STS results when applied with various models, with varying corpora and across multiple languages. Setting a new SOTA score for multiple well established unsupervised English STS tasks.

Additionally, this paper introduced an alteration to the supervised STS regression task proposed by Reimers & Gurevych (2019), which improves the supervised STS scores for all tested models. Using this altered regression task, regular BERT models achieve equally good as when first fine-tuned towards NLI, CT or both. Suggesting that the current Transformer pre-training objectives themselves capture useful sentence level semantic knowledge.

## 8 ACKNOWLEDGEMENTS

This work was partially funded by Vinnova under contract 2019-02996, and the Swedish Foundation for Strategic Research (SSF) under contract RIT15-0046. Finally, the authors wish to thank Joey Öhman, Melker Mossberg and Linus Bein Fahlander, for the spicy Taco evenings which helped us through the rough times of COVID-19.

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

# A APPENDIX

## A.1 VISUAL EXAMPLE OF CONTRASTIVE TENSION

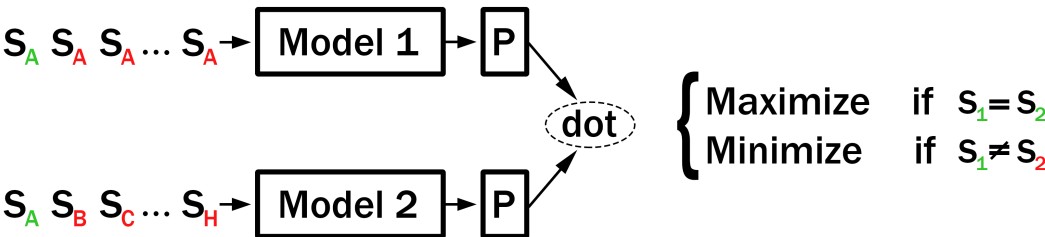

Figure 4: CT follows an architecture similar to a siamese network, but with independent models. Both models are updated based on a contrastive loss on the unnormalized dot product

Figure 4 demonstrates how CT is applied when $K = 7$ which yields 1 positive sentence pair $(S_A, S_A)$ where the models are trained to maximize the dot product, and 7 negative sentence pairs $(S_A, S_B), (S_A, S_C) ... (S_A, S_H)$ where the models are trained to minimize the dot product. As all sentence pairs are included into the same batch, the loss for all $K + 1$ sentence pairs are calculated before updating the models parameters.

## A.2 MODEL NAMING CONVENTION

Throughout the paper we sequentially apply different fine-tuning objectives on pre-trained transformer models, i.e no pre-training is performed during fine-tuning and no two fine-tuning tasks are applied in parallel. The terms *"Distil"*, *"Base"* and *"Large"* are used as descriptors of the pre-trained model's size and in the case of *"Distil"* also its pre-training objective.

The two main fine-tuning tasks we consider is CT and the Siamese NLI task of InferSent and S-BERT. When a model has been tuned towards the Siamese NLI task an additional *"S-"* is added as a prefix to the model base. When a model has been trained with the CT training objective *"-CT"* is added as suffix to the model name. It is strictly the case that when both these tasks are applied to a model, the NLI task is applied before to the CT objective.

For example the model *"S-BERT-Distil-CT"*, is a distilled BERT model that has been fine-tuned towards the S-BERT NLI task, before finally tuned with the CT training objective.

## A.3 LEARNING RATE SCHEDULE

Hyperparameter search concluded that $2e^-6$ was a stable learning rate for applying CT to pre-trained BERT models. However, we found it possible to speedup learning during the early training stages by using a higher learning rate, leading us to the step-wise learning rate schedule seen in table 5. We found no significant difference in the end result when applying the learning rate schedule and when training for a longer period with a smaller learning rate.

Table 5: Step-wise learning schedule applied for all training with Contrastive Tension.

| N #Updates | Learning Rate |
|---|---|
| $N < 500$ | $1e^-5$ |
| $N < 1000$ | $8e^-6$ |
| $N < 1500$ | $6e^-6$ |
| $N < 2000$ | $4e^-6$ |
| $2000 \leq N$ | $2e^-6$ |

### A.4 REPRODUCIBILITY OF PREVIOUS WORK

There exists a discrepancy regarding the reported STS scores between various previous works (Reimers & Gurevych, 2019; Wang & Kuo, 2020). As mentioned, we perform all our STS evaluation with the SentEval framework of Conneau & Kiela (2018), the following are our observations and experiences as we compare with reported results of Reimers & Gurevych (2019), and the follow up work of Wang & Kuo (2020).

For the English **Unsupervised STS** tasks of SemEval 2012-2016 we found that:

1. Our results for BERT, S-BERT and S-BERT-WK are consistent with the work of Wang & Kuo (2020). (Although they do not report results for S-BERT-Large-WK).

2. Reimers & Gurevych (2019) report the highest STS score for the S-BERT models but the lowest for BERT. Which when compared gives an average Spearman difference of 3.9 for S-BERT-Base, 3.4 for S-BERT-Large and $-1.45$ for BERT.

3. No one agrees upon the scores of either InferSent or USE.

When evaluating towards the **Supervised STS** tasks of STS-b test set we found that:

1. Our Spearman results for S-BERT-Base and S-BERT-Large, prior to fine-tuning, are consistent with the work of Reimers & Gurevych (2019). Differing with less than $0.05$ points.

2. Using the official S-BERT code to fine-tune the released S-BERT-Large model, with the described hyperparameters, yields worse results than reported by Reimers & Gurevych (2019), both when evaluating with SentEval or the included evaluation script.

Finally, we note that the official S-BERT implementation continuously evaluates towards the STS-b validation set throughout the NLI training, saving the copy that performs best. This makes all models trained with this setup invalid for the unsupervised STS tasks, as the STS-b validation set is a collection of samples from these tests. This is **not** to say that this is the setup that was used for the results reported by Reimers & Gurevych (2019), but something that future researchers should be aware of.

## B

Additional Experiments and Results

### B.1 ADDITIONAL UNSUPERVISED STS

In order to give further insights on the performance of CT we report unsupervised STS results from 10 CT training runs per considered pre-trained model. Since CT trains two models in parallel this results in 10 CT model pairs i.e. 20 models per considered pre-trained model. Table 6 show the worst/best performing model and the average performance of all 20 models, completely disregarding which models were paired during training. Table 7 showcases how two paired CT models tend to differ in regards to the mean unsupervised STS score, showing the min, max and average difference over the 10 training runs.

It is apparent that out of the models whom have not been tuned towards the NLI task, *BERT-Distil-CT* showcases the most stable performance, with very little variation between the worst and best performing model compared to both *BERT-Base-CT* and *BERT-Large-CT* who show a lower worst possible score. If this discrepancy in performance stability is due to model size or that *BERT-Distil* is trained via distillation is left for future work. Applying CT to a model who has been tuned towards an NLI task seemingly produces both more stable and better results.

Table 6: Pearson and Spearman correlation (x100) on various unsupervised semantic textual similarity tasks.

| | STS12 | STS13 | STS14 | STS15 | STS16 | Avg. |
|---|---|---|---|---|---|---|
| *Worst Performing Model* | | | | | | |
| BERT-Distil-CT | 66.69 / 66.23 | 72.44 / 73.72 | 76.04 / 73.09 | 77.61 / 78.20 | 77.51 / 78.08 | 73.86 / 73.86 |
| BERT-Base-CT | 62.04 / 62.65 | 65.23 / 65.50 | 71.62 / 68.85 | 75.48 / 75.90 | 74.69 / 75.66 | 69.81 / 69.71 |
| BERT-Large-CT | 63.82 / 65.12 | 72.14 / 72.21 | 72.20 / 69.40 | 72.58 / 73.02 | 73.25 / 74.33 | 70.80 / 70.82 |
| S-BERT-Distil-CT | 68.96 / 67.51 | 72.02 / 72.74 | 77.30 / 75.44 | 78.31 / 79.73 | 74.80 / 77.54 | 74.28 / 74.59 |
| S-BERT-Base-CT | 68.09 / 67.69 | 72.54 / 73.35 | 77.25 / 75.80 | 77.93 / 78.99 | 74.59 / 76.61 | 74.08 / 74.49 |
| S-BERT-Large-CT | 70.37 / 68.94 | 74.70 / 74.90 | 78.36 / 76.36 | 79.22 / 80.29 | 74.66 / 77.23 | 75.46 / 75.54 |
| *Best Performing Model* | | | | | | |
| BERT-Distil-CT | 68.14 / 67.22 | 73.48 / 74.04 | 77.03 / 73.45 | 77.88 / 78.56 | 76.54 / 78.15 | 74.61 / 74.28 |
| BERT-Base-CT | 68.20 / 68.56 | 74.33 / 74.50 | 76.76 / 73.33 | 78.71 / 79.29 | 78.10 / 79.15 | 75.22 / 74.97 |
| BERT-Large-CT | 69.26 / 69.03 | 76.90 / **77.19** | 77.71 / 74.50 | 79.08 / 79.58 | **79.16 / 80.07** | 76.42 / 76.07 |
| S-BERT-Distil-CT | 70.04 / 68.25 | 76.78 / 76.98 | 79.69 / 66.68 | 79.66 / 80.37 | 77.54 / 79.63 | 76.74 / 76.58 |
| S-BERT-Base-CT | 69.59 / 68.20 | 74.38 / 75.18 | 78.30 / 76.56 | 79.71 / 80.54 | 75.71 / 77.58 | 75.54 / 75.64 |
| S-BERT-Large-CT | **71.72 / 69.96** | **77.09** / 77.17 | **78.61 / 76.73** | **80.50 / 81.45** | 76.38 / 78.43 | **76.86 / 76.75** |
| *Mean Performance* | | | | | | |
| BERT-Distil-CT | 67.69 / 67.10 | 72.84 / 73.95 | 76.54 / 73.29 | 77.66 / 78.27 | 76.53 / 78.15 | 74.25 / 74.15 |
| BERT-Base-CT | 64.46 / 64.74 | 68.11 / 68.34 | 73.14 / 70.24 | 76.66 / 77.09 | 75.85 / 76.84 | 71.64 / 71.46 |
| BERT-Large-CT | 67.25 / 67.80 | 74.57 / 74.70 | 75.84 / 72.73 | 77.41 / 77.89 | 77.22 / 78.21 | 74.46 / 74.27 |
| S-BERT-Distil-CT | 69.75 / 68.14 | 74.58 / 75.05 | 77.98 / 76.19 | 78.63 / 79.77 | 75.82 / 78.23 | 75.35 / 75.48 |
| S-BERT-Base-CT | 69.10 / 68.38 | 73.61 / 74.36 | 77.90 / 76.27 | 78.88 / 79.88 | 75.06 / 77.11 | 74.91 / 75.20 |
| S-BERT-Large-CT | 71.37 / 69.70 | 75.56 / 75.80 | 78.60 / 77.02 | 79.99 / 80.98 | 75.96 / 78.06 | 76.30 / 76.31 |

Table 7: Min, max and mean difference between CT paired models in Pearson and Spearman correlation (x100) in regards to the mean score of the unsupervised semantic textual similarity tasks.

| | MIN Difference | MAX Difference | MEAN Difference |
|---|---|---|---|
| BERT-Distil-CT | 0.06 / 0.04 | 0.55 / 0.29 | 0.31 / 0.17 |
| BERT-Base-CT | 0.05 / 0.00 | 1.71 / 1.36 | 0.75 / 0.51 |
| BERT-Large-CT | 0.03 / 0.02 | **4.49 / 3.43** | **1.26 / 1.13** |
| S-BERT-Distil-CT | **0.31 / 0.11** | 1.52 / 1.25 | 0.88 / 0.76 |
| S-BERT-Base-CT | 0.23 / 0.09 | 1.03 / 0.90 | 0.58 / 0.33 |
| S-BERT-Large-CT | 0.00 / 0.04 | 0.81 / 0.92 | 0.38 / 0.33 |

## B.2 DOWNSTREAM & PROBING TASKS

To comply with previous work, we evaluate CT on the various set of downstream tasks supplied by the SentEval package (Conneau & Kiela, 2018). As results in table 8 show, we find only minor improvements when using the representations from the fine-tuned models compared to BERT. S-BERT produces a minor improvement for most non semantic related downstream tasks and CT performs slightly better on the semantic related tasks SICK-R and STS-b. Interestingly, the results from table 2 show that BERT-CT, S-BERT and S-BERT-CT all perform better on the STS-b test set when not training an extra linear classifier.

Additionally we evaluate towards the fine grained analysis tasks supplied by SentEval. The results in table 9 clearly show that S-BERT's NLI fine-tuning objective decreases the score in all tests compared to BERT. CT also clearly decreases the performance on all tests except the Bigram Shift task, where this is done to a smaller degree.

Table 8: Results on the downstream tasks supplied with the SentEval package. For the semantic related tasks SICK-R and STS-b, the Pearson correlation (x100) is reported.

| | CR | MR | MPQA | SUBJ | SST2 | SST5 | TREC | MRPC | SICK-E | SICK-R | STS-b | AVG |
|---|---|---|---|---|---|---|---|---|---|---|---|---|
| BERT-Distil | 85.96 | 79.98 | 88.42 | 95.14 | 85.39 | 45.93 | 90.60 | 74.14 | 81.69 | 83.74 | 69.53 | 80.05 |
| BERT-Base | 86.96 | 81.33 | 88.07 | 95.03 | 85.94 | 46.74 | 90.60 | 73.74 | 79.50 | 80.47 | 65.40 | 79.44 |
| BERT-Large | 88.74 | 84.33 | 86.64 | **95.27** | 79.29 | **50.32** | 91.40 | 71.65 | 75.28 | 77.09 | 66.22 | 78.75 |
| BERT-Distil-NLI | 88.37 | 80.83 | 95.50 | 82.54 | 86.99 | 47.47 | 85.60 | 76.12 | 83.15 | 84.72 | 75.90 | 80.11 |
| BERT-Base-NLI | 89.24 | 82.65 | 89.61 | 93.84 | 88.36 | 47.38 | 85.20 | 75.07 | 82.04 | 84.24 | 73.05 | 80.97 |
| BERT-Large-NLI | **90.52** | **84.36** | **90.30** | 94.32 | **90.72** | 50.05 | 86.80 | **76.52** | **83.05** | 84.94 | 75.02 | **82.42** |
| *Our Contributions* | | | | | | | | | | | | |
| BERT-Distil-CT | 84.00 | 78.51 | 88.62 | 93.83 | 83.47 | 45.34 | 87.60 | 74.61 | 81.96 | 85.06 | 74.45 | 79.77 |
| BERT-Base-CT | 84.00 | 79.84 | 88.06 | 94.10 | 82.43 | 45.25 | 73.80 | 80.80 | 84.30 | 73.69 | 79.59 |
| BERT-Large-CT | 86.81 | 82.38 | 88.31 | 94.34 | 87.75 | 46.56 | 88.00 | 73.10 | 81.49 | 84.93 | 76.50 | 80.92 |
| BERT-Distil-NLI-CT | 87.68 | 80.74 | 89.33 | 92.59 | 86.27 | 46.97 | 85.80 | 75.59 | 82.81 | 84.91 | **77.68** | 80.94 |
| BERT-Base-NLI-CT | 88.66 | 81.83 | 89.79 | 93.72 | 87.91 | 47.83 | 83.00 | 74.43 | 82.42 | **85.32** | 77.42 | 81.12 |
| BERT-Large-NLI-CT | 89.56 | 82.56 | 90.20 | 83.08 | 88.85 | 48.60 | 87.20 | 74.43 | 82.77 | 84.88 | 77.49 | 80.87 |

Table 9: Results on the fine grained analysis tasks tasks supplied with the SentEval package.

| | Length | WC | Depth | TopConst | BShift | Tense | SubjNum | ObjNum | OddManOut | CoordInv | AVG |
|---|---|---|---|---|---|---|---|---|---|---|---|
| BERT-Distil | **88.29** | **67.39** | **39.68** | 76.03 | 86.81 | **88.89** | **86.06** | **83.15** | 63.19 | 65.09 | **74.46** |
| BERT-Base | 81.99 | 61.20 | 36.19 | **77.63** | 88.76 | 88.23 | 84.98 | 82.11 | 66.69 | 69.94 | 73.77 |
| BERT-Large | 70.82 | 55.26 | 33.35 | 68.86 | 90.29 | 88.31 | 81.67 | 80.37 | **69.29** | **71.23** | 70.95 |
| BERT-Distil-NLI | 71.23 | 61.76 | 31.83 | 58.75 | 70.41 | 85.11 | 79.02 | 77.71 | 57.93 | 59.10 | 65.29 |
| BERT-Base-NLI | 72.12 | 58.65 | 31.14 | 60.50 | 76.02 | 86.81 | 78.38 | 77.08 | 62.97 | 63.78 | 66.29 |
| BERT-Large-NLI | 59.79 | 54.47 | 29.63 | 57.98 | 76.28 | 84.36 | 76.34 | 73.65 | 64.28 | 65.71 | 64.25 |
| *Our Contributions* | | | | | | | | | | | |
| BERT-Distil-CT | 81.86 | 74.49 | 37.78 | 69.76 | 80.38 | 88.54 | 83.76 | 80.86 | 60.91 | 59.94 | 71.83 |
| BERT-Base-CT | 77.68 | 80.69 | 34.21 | 68.32 | 85.70 | 88.03 | 83.70 | 80.35 | 64.93 | 64.97 | 72.86 |
| BERT-Large-CT | 64.85 | 65.93 | 30.97 | 64.75 | **86.59** | 87.9 | 81.17 | 80.85 | 68.04 | 67.37 | 69.84 |
| BERT-Distil-NLI-CT | 72.55 | 67.84 | 33.29 | 62.61 | 72.49 | 86.20 | 82.01 | 78.84 | 58.40 | 59.61 | 67.38 |
| BERT-Base-NLI-CT | 71.44 | 66.42 | 32.19 | 61.45 | 77.61 | 87.87 | 80.10 | 78.45 | 63.21 | 63.83 | 68.26 |
| BERT-Large-NLI-CT | 59.26 | 64.85 | 29.11 | 58.54 | 75.79 | 83.05 | 76.98 | 74.17 | 62.88 | 63.63 | 64.83 |

## B.3    GLUE BENCHMARK

We evaluate our BERT-Base-CT model on the General Language Understanding Evaluation (GLUE) benchmark (Wang et al., 2019), and compare with BERT-Base and S-BERT-Base. Following Devlin et al. (2019), we chose the best performing model on the validation set for each combination of learning rate (among 5e-5, 4e-5, 3e-5, 2e-5 for BERT-base and among 5e-5, 4e-5, 3e-5, 2e-5, 1e-5, 2e-6 for the other models), model and task. For all GLUE tasks, all models are fine-tuned using a batch size of 32 for three epochs.

The results presented in Table 10 demonstrates that BERT performs the best, with a slight margin, on near all tasks. Exception being the STS-b task, where both S-BERT and BERT-CT see a slight improvement over BERT. However, as depicted in Table 2 both S-BERT and BERT-CT attain higher test scores with the embedding based approach, compared to feeding both sentences to the same model as is done in the GLUE tasks.

Table 10:    GLUE Test results, returned by the GLUE evaluation server (`https://gluebenchmark.com/leaderboard`). Following Devlin et al. (2019), the WNLI set has been excluded from the computation of the average score. F1 score is reported for QQP and MRPC, Spearman correlation (x100) for STS-b and accuracy is reported for the rest of the tasks.

|  | MNLI-(m/mm) | QQP | QNLI | SST-2 | CoLA | STS-b | MRPC | RTE | Average |
|---|---|---|---|---|---|---|---|---|---|
| BERT-Base | **84.2/83.6** | **71.3** | **90.6** | **91.7** | **51.9** | 83.6 | **87.8** | **65.0** | **78.8** |
| S-BERT-Base | 83.9/83.1 | **71.3** | 90.5 | 90.9 | 47.0 | **84.7** | 85.3 | 61.6 | 77.6 |
| BERT-Base-CT | 82.3/81.9 | 70.1 | 89.7 | 91.3 | 48.8 | 84.0 | 84.4 | 61.1 | 77.0 |

## B.4   LAYER-WISE STS STUDY OF COMMON TRANSFORMER MODELS

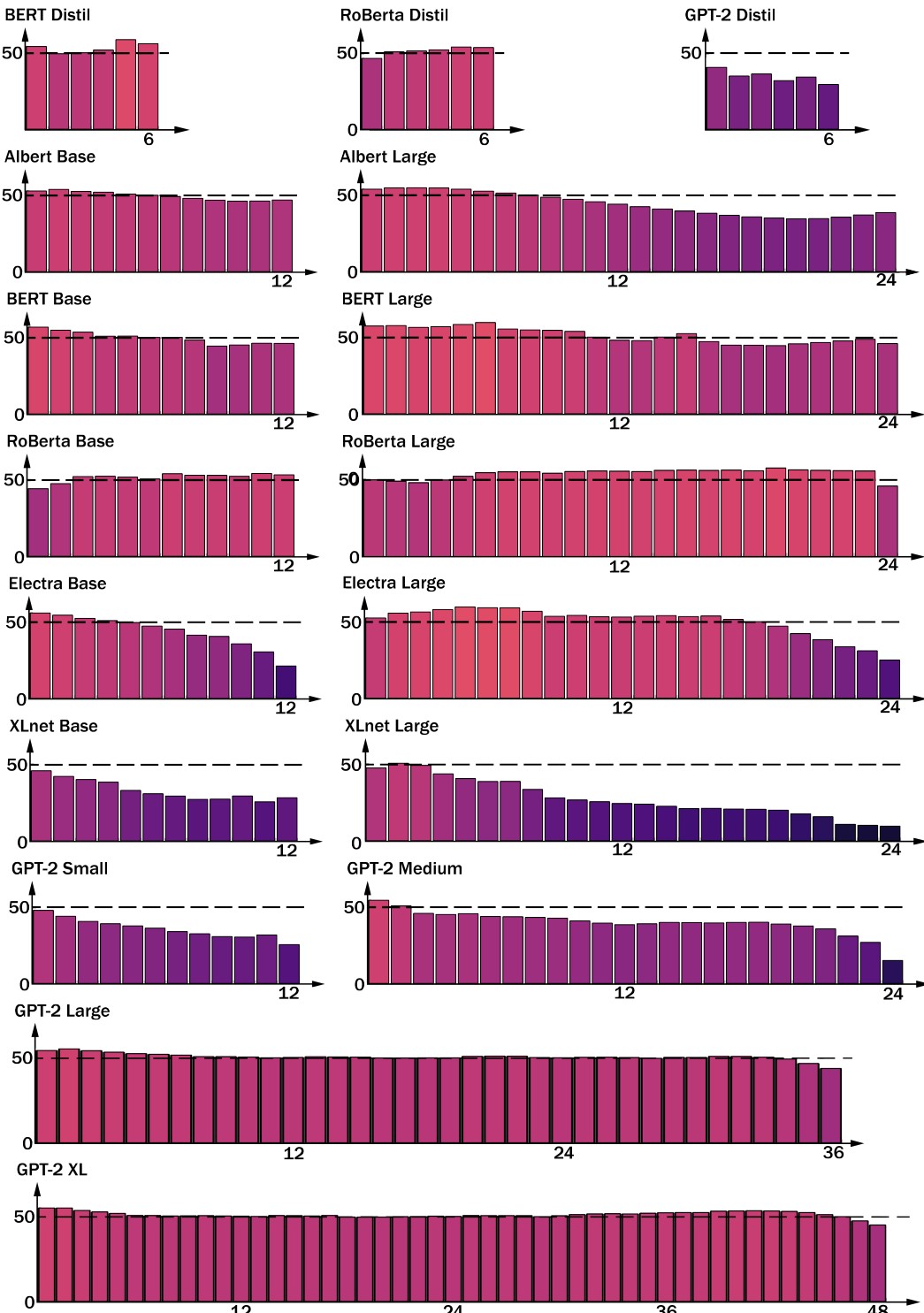

Figure 5: Layer-wise unsupervised STS performance on the STS-b test set. X-axis denotes the layers of the depicted model and the Y-axis denotes the Spearman correlation (x100). Color is a redundant indicator of the Spearman correlation and the value 50 is included for visual comparison.

## B.5  Layer-wise STS Study before and after fine-tuning

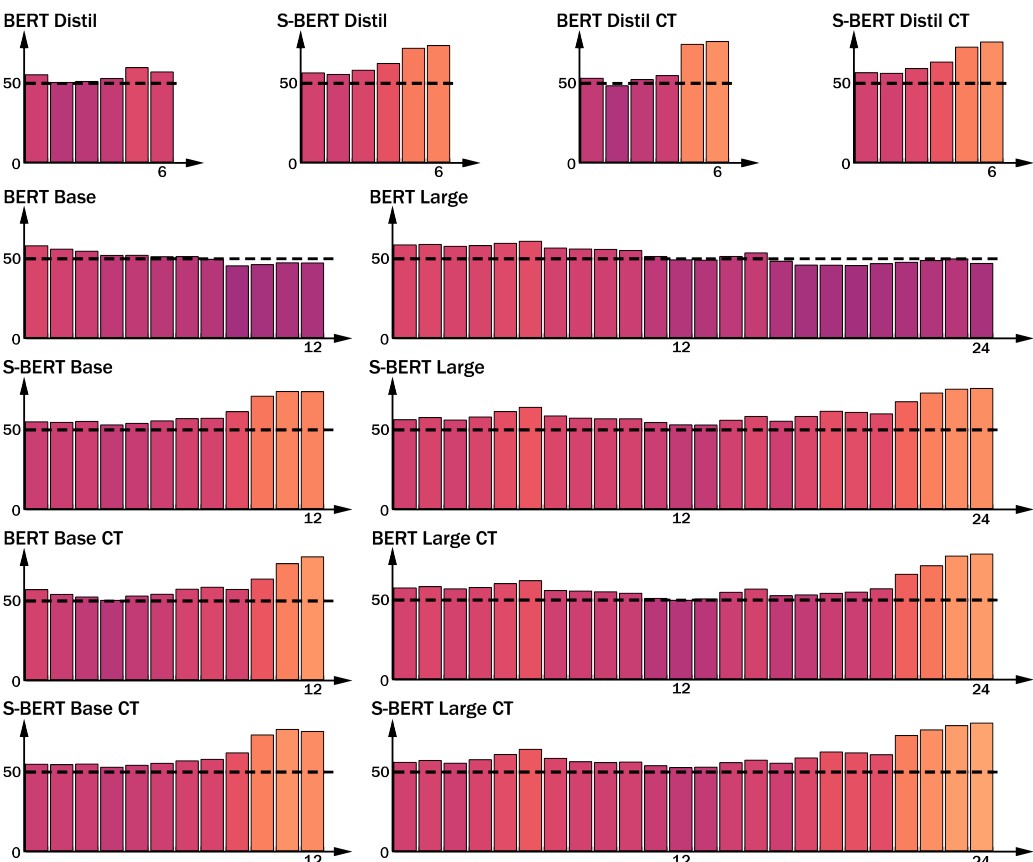

Figure 6: Layer-wise unsupervised STS performance on the STS-b test set. X-axis denotes the layers of the depicted model and the Y-axis denotes the Spearman correlation (x100). Color is a redundant indicator of the Spearman correlation and the value 50 is included for visual comparison.

## C  EXPERIMENT SETUP

### C.1  MODEL CHECKPOINTS

All models and checkpoints where implemented and loaded using the Huggingface API (Wolf et al., 2019). The various model checkpoints used throughout the experiments of this paper are available in Table 11.

Table 11: Model checkpoints used in these experiments.

| Model Name | Parameters | URL |
|---|---|---|
| **English Bert Models** | | |
| Bert-Distil | 66 M | *huggingface.co/distilbert-base-uncased* |
| Bert-Base | 110 M | *huggingface.co/bert-base-uncased* |
| Bert-Large | 340 M | *huggingface.co/distilbert-base-uncased* |
| S-Bert-Distil | 66 M | *Anonymous Upload* |
| S-Bert-Base | 110 M | *https://huggingface.co/sentence-transformers/bert-base-nli-mean-tokens* |
| S-Bert-Large | 340 M | *https://huggingface.co/sentence-transformers/bert-large-nli-mean-tokens* |
| **Multilingual Models** | | |
| Arabic Bert-Base | 110 M | *huggingface.co/asafaya/bert-base-arabic* |
| Spanish Bert-Base | 110 M | *https://huggingface.co/dccuchile/bert-base-spanish-wwm-uncased* |
| Swedish Bert-Base | 110 M | *https://huggingface.co/KB/bert-base-swedish-cased* |
| Russian Bert-Base | 110 M | *https://huggingface.co/DeepPavlov/rubert-base-cased* |
| Multilingual Bert-Base | 110 M | *https://huggingface.co/bert-base-multilingual-cased* |
| XLM-R | 571 M | *https://huggingface.co/xlm-mlm-100-1280* |
| **Additional Models** | | |
| Albert-Base | 11 M | *https://huggingface.co/albert-base-v1* |
| Albert-Large | 17 M | *https://huggingface.co/albert-large-v1* |
| Electra-Base | 110 M | *https://huggingface.co/google/electra-base-discriminator* |
| Electra-Large | 340 M | *https://huggingface.co/google/electra-large-discriminator* |
| GPT2-Small | 117 M | *https://huggingface.co/gpt2* |
| GPT2-Medium | 345 M | *https://huggingface.co/gpt2-medium* |
| GPT2-Large | 774 M | *https://huggingface.co/gpt2-large* |
| GPT2-XL | 1558 M | *https://huggingface.co/gpt2-xl* |
| RoBerta-Base | 125 M | *https://huggingface.co/roberta-base* |
| RoBerta-Large | 355 M | *https://huggingface.co/roberta-large* |
| XLNet-Base | 110 M | *https://huggingface.co/xlnet-base-cased* |
| XLNet-Large | 340 M | *https://huggingface.co/xlnet-large-cased* |

### C.2  WIKIPEDIA DUMPS

All Wikipedia text data was pre-processed using the WikiExtractor provided by Attardi (2015). The dump files used as text corpora throughout the experiments of this paper are available in Table 12.

Table 12: Wikipedia dumps used in these experiments.

| Language | URL |
|---|---|
| Arabic | https://dumps.wikimedia.org/arwiki/20200820/arwiki-20200820-pages-articles-multistream.xml.bz2 |
| English | https://dumps.wikimedia.org/enwiki/20200820/enwiki-20200820-pages-articles-multistream.xml.bz2 |
| Russian | https://dumps.wikimedia.org/ruwiki/20200820/ruwiki-20200820-pages-articles-multistream.xml.bz2 |
| Spanish | https://dumps.wikimedia.org/eswiki/20200820/eswiki-20200820-pages-articles-multistream.xml.bz2 |
| Swedish | https://dumps.wikimedia.org/svwiki/20200820/svwiki-20200820-pages-articles-multistream.xml.bz2 |

