# OpenReview forum: "Semantic Re-tuning with Contrastive Tension"
_ICLR.cc/2021/Conference — ICLR 2021 Poster_

### Official Review · AnonReviewer4 · 2020-10-28
**Interesting observation with simple method on text representation learning for STS**

**Rating:** 7
**Confidence:** 5

**Review:**

The paper studies the problem of finding effective representation for semantic text similarity (STS). The paper first investigates the effectiveness of pre-trained masked language models (e.g. BERT) in the STS task. They found different layers of BERT have different performance when employed in STS task -- in particular, the popular method of using the last layer does not usually lead to  good performance. In fact, the last layers are worse than those preceding layers in STS. This is universal in several models including BERT, Electra, XLNet, and GPT-2, but not RoBerta. The paper then propose to use dual contrastive training method (basically use two BERT branches) to further fine-tune BERT, where the additional objective is defined by bringing two models' output closer for the same input sentence and further for randomly sampled different sentences.
The authors evaluated their method on standard benchmark of STS task and obtain consistent improvement over BERT (unsupervised setting) and S-BERT (supervised setting)

Strong points of the paper:
S1. The observation of earlier layers of pre-trained masked language models work better than later layers on STS is interesting and surprising. It is consistent in all models except Roberta-base.
S2. The proposed method is relatively simple to implement.
S3. The experimental results on STS task verifies that the proposed method achieves consistent and significant improvement over the plain BERT and supervised BERT.

Weak points:
W1. Although the observation is interesting, the connection between the observation on layer-wise performance of BERT on STS and the proposed contrastive training method is not clear.
W2. The paper lacks details in the training of BERT with CT and BERT-Distil. It is also unclear whether the BERT-base-CT is using BERT-base as the initialization.
W3. Though the method works for semantic text similarity, it does not work for other NLP tasks in particular on GLUE. It is unclear why.

Additional comments and questions:
Since it only works on STS, the title of the paper may appear too broad. The authors may consider revising the title.
Some statements in the introduction are not backed by evidence. For example, the layers close to the final objective function will be more task-specific.
In the abstract, it states "impose a significant bias on the the final layers". What bias?
The authors may want to avoid using too long compound sentences, which hurts readability.
Figure 1. what is the range of y-axis?

---

> ### Author Response · Authors · 2020-11-11
> **A solid review providing fair feedback**
>
> Thank you very much for your effort, the feedback you raise is valuable to us. Some of the questions and ambiguities you found are addressed in the General Response.
>
> ##### Regarding the comments about the introduction and abstract:
> We will provide references supporting our claims, for example “features residing in layers close to the objective function will be task-specific.” in the revised version of the paper. Regarding the use of the term “bias”, by bias we mean that the final layers encode information specific to the masked language modeling task.
>
> ##### Regarding the performance on non-STS taks.
> For the GLUE tasks, we used the standard BERT-finetuning approach, using the CT-model as an initialization (We will clarify this in appendix B.2). As to why the results are not as impressive as the STS-results, we believe this has to do with the fact that the CT-objective is, in a sense, non-constructive: No new information is added to the model during CT training, but realigns the pre-trained model such that the output encodes sentence-level semantics. As a result of this, we do not expect a CT initialized model to fare better in a supervised sentence-pair task, given ample training data.
>
> For the non-STS tasks on senteval, we observe a general trend that the STS-scores of sentence representations are not necessarily correlated with the score on other tasks: While BERT-Large is the least performant model on SICK-R and STS-B (both sentence-level semantic similarity tasks), it is the most performant on a number of other tasks (SUBJ, SST5, TREC). We do not know why this is. Also, the discrepancy between section 5.2 and the Senteval version of STS-B that includes training a linear model (we fare much better without the linear model) casts doubt on the way Senteval trains its linear model for the regression tasks.

---

### Official Review · AnonReviewer1 · 2020-10-29
**Weak accept**

**Rating:** 6
**Confidence:** 4

**Review:**

This paper proposes Contrastive Tension, a self-supervised method to improve sentence representations from pre-trained language models for Semantic Textual Similarity tasks. This work is motivated by the observation by previous work that the final layers of pre-trained model are often biased towards token-level pre-training objectives, and perform poorly for sentence similarity tasks. The proposed method counters this bias by introducing a sentence-level self-supervised task where two different models are encouraged to generate similar representation for the same input sentence, and different representations for different inputs. Experiments show the proposed method significantly improves over previous SotA methods on STS benchmarks.

Pros:
1) The proposed method only relies on unlabeled data.
2) The experimental results on STS benchmarks are strong. The authors also emphasizes reproducibility by reporting lowest performing numbers and promise to release code/model, which is a plus.
Cons:
1) The training objective involving two separate models is not clearly motivated. Since the two models are initialized with identical weights from pre-trained models, the loss of contrastive tension should be very small at the start of fine-tuning. More analysis on the training process, such as training loss curve, could be helpful in this regard.
2) The paper propose to improve the extraction of representations from pre-trained models, but the experiment is only done on STS datasets. I would like to see results on other tasks such MNLI where the sentence level representations are used.

Overall, I find the paper interesting, but the applicability of the method is not clearly demonstrated since the experiment is only done on STS datasets. Thus, I give this paper a weak accept rating.

---

> ### Author Response · Authors · 2020-11-11
> **Good feedback mixed with some misaligned intuition**
>
> Your review and feedback is much appreciated, thank you! We do however hope we can convince you further and clarify certain issues. See the general response for explanations regarding the core idea of CT and its loss.
>
>
> ##### Regarding results on other tasks:
> The main presentation of the results are on unsupervised STS benchmarks. We do however report senteval-benchmarks (sentence representation tasks) in appendix B.1 table 6-7, where SICK-E is an NLI-task.
>
> We also report GLUE benchmarks in appendix B.2 table 8 (note that the GLUE benchmarks are not based on the sentence representations, but the whole BERT-CT-model, for more discussion regarding this, see response to reviewer #4.).
>
> ##### Regarding the applicability of Sentence Representations:
> We disagree with the sentiment that the applicability of the method is in question since we only evaluate on STS. Unsupervised methods for training sentence representations are an important part of computational semantics.  Immediate application areas are for example: information retrieval, semantic search etc..

---

### Official Review · AnonReviewer2 · 2020-10-30
**Interesting idea but some issues with the explanation and evaluation**

**Rating:** 5
**Confidence:** 4

**Review:**

The paper describes a method for improving pre-trained language representations for sentence similarity tasks.
During the training, two independent models, with identically initialized weights, are trained to maximise the dot product between their sentence representations for identical sentences.
This is shown to improve performance on different sentence similarity benchmarks.

The method could potentially be useful and shows positive empirical results. But there are several questionable aspects that are not addressed in the paper.

The evaluation is currently not convincing. SOTA results are claimed on STS 2012-2016 datasets, but very few contemporary models have been reported on those datasets. Results on the much more widely used STS-B are considerably lower than existing SOTA. Many models, including XLNet, RoBERTa and ALBERT report over 92%, whereas the proposed model achieves 86% on STS-B.

The core of the idea is that the output of two identical models is optimized to be similar to each other. As far as I can tell based on the paper, the only difference then is that negative samples are always passed through one particular model. Without this, the models would remain identical throughout training and there would be no point in optimizing the representations to be similar. It should be more clearly motivated and investigated why this difference would provide any benefit to the model representations.

The method should be compared to other methods that use unsupervised sentence-level objectives for learning sentence representations. QuickThoughts and DeCLUTR are mentioned in the related work section but not compared.

Citations for BERT, Electra, XLNet and GPT-2 are missing in Section 3.

In section 3 it is demonstrated that lower or intermediate layers are generally much more competitive on sentence similarity tasks, compared to the top layer of transformer models. However, it seems that the proposed CT method is only applied to and compared to representations from the top layer of the models. Why not start with a representation that already performs better? At least for the baselines the performance of the best layer should be reported, as simply taking a different layer from the pre-trained model would be a much simpler approach.

The paper says that the worst-performing model is reported. Why? That seems very unusual. It would be more informative to report min and max results or the average.
But also worst among what? Only one model configuration is described, so what are the other (better?) models?

In the supervised setting it is unclear how the experiments are set up. Does CT training happen before, after or parallel with the supervised training?

---

> ### Author Response · Authors · 2020-11-11
> **A response hoping to clarify certain misunderstandings**
>
> Thank you for your review and feedback! There seems however to be certain things that we have failed to convey and convince you of. In addition to this response, we therefore hope you find our explanation in the general response to clarify some of these matters.
>
> ##### Regarding the Unsupervised STS results:
> We agree that including results for additional related work would strengthen the result sections. Comparing CT with older self-supervised algorithms like Skip-thoughts, Quick-Thoughts and potentially DeCLUTR, could be very insightful.
>
> These algorithms were not included due to spatial reasons and that we had a hard time finding good pre-trained models for these older algorithms. As stated in the paper, we found a discrepancy between previously reported results, so we only included results for the models we could evaluate ourselves. We felt content as we managed to include InferSent, S-BERT and the most recent USE, which are the most competitive and recent predecessors.
> Neither Skip-thoughts or Quick-thoughts have been the SOTA for quite some time.
>
> DeCLUTR was skipped since it is yet to be officially peer-reviewed (to our knowledge), and we find some of their reported results to be questionable. But we can be convinced to include DeCLUTR if required. Although, DeCLUTR themselves don’t report any SOTA scores, hence it is our understanding that CT is in fact the SOTA for unsupervised STS.
>
> ##### Regarding the Supervised STS results:
> I am afraid that you have misinterpreted this section. While it is true that the models you mention all report STS-b results well above 90%, this is for a pairwise-comparison of sentences. We might have been unclear on this point, but this evaluation section, as the rest of the paper, is only concerned with evaluation of sentence embeddings (With the exception of GLUE evaluation in the appendix). The previous SOTA score for STS-b using sentence representation is S-BERT, who reports a Spearman score of 86.10(+- 0.13) for S-BERT Large and  86.13(+- 0.35) for S-RoBerta Large. We will gladly make rearrangements to this section to clarify this.
>
> The main point we wished to convey in this section is the importance of taking the cosine spread into consideration prior to fine-tuning. As demonstrated with the relative regression targets, we found a clear performance increase when accounting for this which resulted in a Spearman score of 86.43 for S-BERT-CT Large.
>
> ##### Regarding missing Citations in section 3:
> Thank you for pointing this out! We agree that although we cited these 4 models in the introduction we did so without explicitly naming them. Do you agree that a more explicit citation and naming in the introduction is sufficient?
>
> ##### Regarding STS in intermediate and early layers:
> Including the best performing layer from a model such as BERT is a great idea!If this does not cause to many spatial problems, we will include this result in the relevant tables.
>
> We fully agree with your intuition that applying training objectives to intermediate layers is an interesting avenue to pursue. However, considering the paper limit imposed by ICLR we would argue that this belongs in a future work or discussion section.
>
>
> ##### Regarding ambiguity in supervised STS:
> Thank you for mentioning this ambiguity. This is something we would gladly clarify with a suitable alteration. For all these experiments we only apply the supervised regression task afterwards the other training tasks has been applied. Meaning that we fine-tune towards STS from either a pretrained BERT, S-BERT or either of the two, after CT has been applied to it.

---

> > ### Comment · ~John_Michael_Giorgi1 · 2021-01-29
> > **DeCLUTR results being questionable**
> >
> > Hi,
> >
> > Great paper, I really enjoyed reading it.
> >
> > I am one of the authors from DeCLUTR. May I ask which of our results specifically you are finding questionable? We would like to clear up any confusion or fix any mistakes you believe you have found in our evaluation.
> >
> > Thanks!

---

> > > ### Comment · ~Fredrik_Carlsson1 · 2021-01-29
> > > **DeCLUTR Results**
> > >
> > > Hey,
> > > Glad that you're reaching out, and thanks for the kind words!
> > >
> > > As I wrote in the review response, I'm uncertain if DeCLUTR has passed any peer-review process, so please correct me if I'm wrong on this! In this paper we're mainly concerned with Semantic Similarity, hence the results which we're a bit confused over is therefore your reported STS scores in Table S3.
> > >
> > > In particular, your reported Sentence Transformers(S-Bert) have scores lower than that we found, and significantly lower than that reported in the original S-Bert paper. Your reported USE score is quite different from both the ones that we reproduced and previously reported. In general, the scores for older models such as InferSent also differ from ours, but this might be more understandable since they're harder to get your hands on than the Huggingface available transformers.
> > >
> > > Now, we have ourselves had problems reproducing results of previous methods, and we have a short discussion concerning this in Appendix-A4. It is a bit heart-wrenching that a lot of papers published in this area all report different results for different models... Especially since it's becoming easier to make your models easily accessible to the public.
> > > I can't vouch for the validity of SentEval, but it's clear that some kind of unified testing environment is required if we're to mitigate this confusion.
> > >
> > > Hope this answers your question.
> > > Let me know if you have any more thoughts on this, and good luck with DeCLUTR and potential improvements :)

---

### Official Review · AnonReviewer3 · 2020-11-10
**Contrastive Tension (CT) credibly improves performance on unsupervised STS**

**Rating:** 9
**Confidence:** 5

**Review:**

The paper investigates a new training objective, contrastive tension (CT), for obtaining unsupervised sentence embeddings. The objective operates by initializing two models with identical weights and then training the models to produce similar sentence embeddings to each other for identical sentences and dissimilar representations for different sentences. This objective encourages the paired models to agree on positive examples, but at the same time encourages divergence in their models weights by providing different sentences to each encoders for the negative pairs containing different sentences. The new objective is applied as an unsupervised finetuning tasks for BERT, Sentence-BERT, Distill BERT, multilingual BERT, XLNet and XLMR.

The paper demonstrates consistently strong results empirical results on the unsupervised semantic textual similarity (STS) task. The results on supervised STS are mixed with more modest gains and losses over the baseline for some configurations. The paper provides reasonably good analysis on demonstrating the impact of the proposed technique within different layers of a pre-trained model as well as on the effect on the model scores vs. human labels. I found the analysis particularly interesting that showed CT helped the model to better discriminate/score pairs with lower similarity scores (Figure 3). I also like the breadth of the experiments that included a number of different models, finetuning corpora and STS datasets includes multilingual STS.

In terms of potential improvements, I found the results in Figure 1 somewhat surprising in that they show that without further fine-tuning on either unsupervised or supervised data the default representations for many pre-training models are poorly suited for STS. It is well established that the final layers are not useful, but I haven't yet seen an analysis looking at all the layers of so many models. The presentation could possible be improved by including for contrast one existing transformer model that performs well on the STS task (e.g., S-BERT or maybe USE).

While presenting the worst-performing results across runs is a refreshing change from other papers that might be cherry picking their results, I found the inclusion of just the worst-performing results makes it a little hard to full understand the performance of the model. If possible update all of the results with the worst-performance across runs with average and worst, maybe using something like AvgScore[WorseScore]. If this makes the tables too crowded, consider including more comprehensive results in the appendix of the paper.

---

> ### Author Response · Authors · 2020-11-11
> **A grateful response**
>
> Thank you for your rigorous effort! It gladdens us that you seem to assign as much potential and importance to our contribution as we do. Your explanation regarding how CT operates is well aligned with ours.  We will keep your feedback regarding the presentation of our contribution well in mind when we revise the paper.
>
> ##### On including layer-wise visualizations of e.g. S-BERT.
> This would certainly be an informative and useful contrast. We omitted it in the current version of the paper due to space limitations. But are open to including it.
>
> ##### Regarding the reporting of the worst-performing model.
> See the general response.

---

### Author Response · Authors · 2020-11-11
**General Response to official reviews**

We are deeply grateful for the thorough work put into the reviews and well formulated feedback.

##### General changes to the paper:
* The title of the paper will be replaced with a more suitable one.
* Equation (1) currently contains a typo that will be fixed.
* Certain tables have misplaced bold texts that will be fixed.
* The different axes and figure captions will be clarified.
* In order to dispel the confusion about our model terminology, we are going to standardize model names and elaborate upon our naming convention.

##### Regarding the core idea of CT:
There seems to be some confusion regarding exactly how and why CT works. Hence we plan to polish this explanation in the paper, meanwhile we provide some brief explanations here:

While it’s true that the two models are trained to maximize the dot product between identical sentences, it’s important to remember that they are also trained to minimize the dot product between the two models' representations for differing sentences. As demonstrated in Figure 3, all the sentences are fairly similar before any fine-tuning. This entails that we initially have a very high loss, as the models dissipate their embedding spaces.
The “Tension” in Contrastive Tension hence originates from the models having to balance these tasks during training. As displayed in figure 3, this objective allows the models to retain similar representations for semantically similar sentences, while increasing the distance between non-semantically similar sentences.

Our preference is that the method sections should be lean and straight-forward, so the plan is to elaborate upon this further in the discussion section. We would also like to clarify that we don’t claim to provide a concrete theoretical proof of why CT works as well as it does. Rather, our arguments are mainly based on the observation regarding the Cosine-Similarity spread of the embeddings spaces, combined with our general intuitions.


##### Regarding the worst performing model being reported in unsupervised STS:
As CT trains two independent models simultaneously, they will most likely reach different performance levels. Since we’re interested in unsupervised STS we have no way to compare the two CT models, in contrast to the supervised setting where a Dev-set can be utilized. When we say the “worst” model, we are referring to the worst model of these two CT models. Reporting the worst of these two models therefore provides a “worst case” scenario for that CT run.

We fully agree that including more informative results for unsupervised STS, with min, max, avg, is a good idea. This was not included in the submission as it makes the tables crowded. We’re aiming to follow the proposed fix of Reviewer-3 by including such results in the appendix, and adding a relevant pointer in the text.

---

### Decision · Program_Chairs · 2021-01-07
**Final Decision**

**Decision:**

Accept (Poster)

**Comment:**


This paper described a model that improves the performance of LM-based pre-trained sentence representation on semantic text similarity tasks (STS). The proposed approach is motivated by the observation that top-layers in transformer-based LMs are quite poor at this task per se. This paper proposes Contrastive Tension, a self-supervised objective that drags representations of same sentence together, and pulls away representations of different sentences. The proposed method only relies on unlabelled data, and is relatively simple to implement. The paper demonstrates consistently strong results empirical results on the unsupervised semantic textual similarity (STS) task. Moreover, the paper provides reasonably good analysis.

On a negative side, the reviewers noted that the paper lacks a bit of analysis about the objective. The connection between the observation on layer-wise performance of BERT on STS and the proposed contrastive training method is not clear. Second, while the result is interesting, its applicability is limited to STS.

Taking into account all the above, the reviews constitue a case for a solid weak-accept. Therefore I recommend acceptance as a Poster contribution.